# PVPO: Pre-Estimated Value-Based Policy Optimization for Agentic Reasoning

## Abstract

Critic-free reinforcement learning methods, particularly group policies, have attracted considerable attention for their efficiency in complex tasks. However, these methods rely heavily on multiple sampling and comparisons within the policy to estimate advantage, which may cause the policy to fall into local optimum and increase computational cost. To address these issues, we propose PVPO, an efficient reinforcement learning method enhanced by an advantage reference anchor and data pre-sampling. Specifically, we use the reference model to rollout in advance and employ the calculated reward score as a reference anchor. Our approach effectively corrects the cumulative bias introduced by intra-group comparisons and significantly reduces reliance on the number of rollouts during training. Meanwhile, the reference model can assess sample difficulty during data pre-sampling, enabling effective selection of high-gain data to improve training efficiency. Moreover, PVPO is orthogonal to other advanced critic-free RL algorithms, making it compatible with and complementary to these methods. Experiments conducted on nine datasets across two domains demonstrate that PVPO achieves State-Of-The-Art (SOTA) performance. Our approach not only demonstrates robust generalization across multiple tasks, but also exhibits scalable performance across models of varying scales.

## 1 Introduction

Reinforcement Learning (RL) is a machine learning method for learning optimal policies through interaction with the environment. Policy optimization depends on accurately estimating the advantage function to improve the agent's actions. In classic actor-critic frameworks, a critic network predicts state-value ($V$), which combines with action-value ($Q$) to compute the advantage and then guides policy updates. Recently, research has increasingly focused on more efficient critic-free architectures. These methods do not directly compute the absolute advantage. Instead, they build baselines for relative advantage, simplifying the training process and reducing resource consumption (Shao et al., 2024; Feng et al., 2025b).

Grouping policies, as used in critic-free RL methods like GRPO (Shao et al., 2024), become an important research topic. This is not only because they demonstrate superior performance, but also because the removal of the value model saves training resources, enabling researchers to train larger-scale models under limited hardware conditions. Although PPO and other actor-critic methods sometimes achieve higher accuracy, critic-free grouping policies are widely used for their practical efficiency. Some studies group by sample, running multiple trajectories within each group to compute relative advantage (Zuo et al., 2025; Lyu et al., 2025). Others group by action or timestep, enabling finer partitioning and more accurate baseline estimation (Feng et al., 2025b; Li et al., 2025a). These methods can improve baseline accuracy for similar trajectories. However, grouping policies usually require more rollouts to boost performance, which greatly increases computational cost. Methods such as DAPO (Yu et al., 2025) aim to mitigate this issue by prioritizing high-value data sampling. However, they primarily redistribute resource utilization rather than achieving a genuine reduction in overall resource consumption. We still need to achieve an effective trade-off between training performance and computational cost. To construct the relative advantage, some methods use state-independent baselines to generate advantage values for each action (Williams, 1992; Ahmadian et al., 2024). GRPO (Shao et al., 2024) and GiGPO (Feng et al., 2025b) compare the rewards of actions or trajectories within groups. In these approaches, the evaluation criterion is derived from the

policy itself, which may cause policy optimization to become confined to existing behavior patterns and lead to local optima.

From a human learning perspective, rollout can be seen as repeated practice. Grouping policies resemble trial-and-error learning, where individuals often compare outcomes to a fixed **Reference Anchor** for more efficient learning. This anchor serves as an objective reference point, distinct from the idealized optimal solutions provided by a critic or the dynamic relative performance within a group, and establishes a more general advantage baseline.

In this paper, we introduce Pre-estimated Value-based Policy Optimization (PVPO), a generalized RL method based on Proximal Policy Optimization (PPO) (Schulman et al., 2017). PVPO adopts a critic-free architecture, is compatible with mainstream group policy RL methods, and maintains low computational cost for grouping, thus effectively combining the strengths of both approaches. Specifically, we use a Reference Model (Ref) to run grouping reasoning and calculate a task-based reward score as an anchor. This anchor serves as the $V$ estimate during RL training, helping to correct the cumulative bias in relative advantage calculations typically observed in large language models (LLMs). In essence, our method decouples $Q$ and $V$ in the grouping policy advantage calculation. The reference anchor is computed in an unsupervised manner and acts as both a supplement and an enhancement to the training dataset, without incurring additional time or memory overhead. In summary, our core contributions are as follows.

- We propose PVPO, an efficient and generalizable approach to critic-free reinforcement learning. PVPO provides a stable, low-variance, and globally consistent advantage function, effectively mitigating concerns of error accumulation and policy drift during training. As a result, PVPO enables more efficient and robust policy optimization while significantly reducing spatio-temporal overhead.

- We introduce a group sampling strategy that offline filters data with unstable accuracy rates to construct high-quality batches, thereby enhancing convergence and learning efficiency. Furthermore, for samples with zero accuracy (i.e., zero reward), we leverage a large-scale LLM to generate ground-truth trajectories, facilitating more effective learning from sparse reward signals.

- PVPO achieves state-of-the-art performance on multi-step retrieval datasets and demonstrates strong generalization on mathematical reasoning benchmarks. Experimental results indicate that PVPO not only enhances multi-hop question answering (QA) and tool-use capabilities, but also improves the overall reasoning ability of LLMs.

## 2 RELATED WORK

### 2.1 AGENTIC REASONING

Leveraging reinforcement learning to drive search represents an important direction in agentic reasoning (Jin et al., 2025; Jiang et al., 2025). Search-o1 (Li et al., 2025b) integrates an agentic search workflow into the reasoning trajectory. This achieves an elegant integration of search and reasoning, sparking a wave of subsequent optimizations (Qian et al., 2025; Wang et al., 2025; Feng et al., 2025a). Moreover, numerous studies on Retrieval-Augmented Generation (RAG) (Li et al., 2025b; Feng et al., 2025c; Hao et al., 2025) have advanced the capabilities of LLM in tool use and information retrieval.

However, existing studies often directly apply algorithms such as GRPO, which are intrinsically ill-suited to the sparse-reward setting of agentic search. These methods depend on dense token-level rewards, necessitating extensive rollouts to achieve stable advantage estimation. Consequently, the quality of the learning signal becomes tightly coupled with the sample size. Our PVPO framework is tailored for agentic search by decoupling the advantage function ($A=Q$-$V$), thereby mitigating sample size dependency. While the actual return ($Q$) leverages the sample size, the advantage baseline ($V$) remains independent of both the current and previous policies. This design ensures a stable learning signal even under severe reward sparsity (e.g., $Q=0$), obviating the need for extensive rollouts.

## 2.2 RL AND LLMs

Recently, reward and advantage computation has been redefined through dynamic generation and iterative optimization, substantially enhancing the performance of critic-free RL methods. Some methods construct denser feedback signals by increasing reward frequency (Bensal et al., 2025; Chen et al., 2024), while others improve reward adherence via extra training phases (Dong et al., 2025). These approaches, however, often suffer from high computational cost and instability rooted in repeated online sampling, drifting rollout distributions, and cumulative estimation bias.

Another line of research aims to recover endogenous rewards from the actor model via reverse engineering (Li et al., 2025c; Zhao et al., 2025), removing the need for extra training and enabling prompt-based adaptation to various evaluation criteria. However, the effectiveness of these rewards is limited by the base model's quality, and reliably guiding reward signals through prompting remains challenging (Zhao et al., 2021; Lu et al., 2022; Liu et al., 2023). To address these challenges, static methods such as offline RL (Kumar et al., 2020; Kostrikov et al., 2022), Direct Preference Optimization (Rafailov et al., 2023; Ethayarajh et al., 2024), and weighted behavioral cloning (Xu et al., 2022a;b) have been investigated, but often trade off adaptability for efficiency due to simple or static estimation of advantage.

Recent research emphasizes that efficient and robust RL depend on adaptive sampling, static baselines, and data-driven estimator selection. Corrado & Hanna (2024) shows that matching empirical experience distributions to the policy significantly improves sample efficiency. Hanna et al. (2019) demonstrates that using empirically estimated behavior policies yields lower-variance evaluation. Q-Prop (Gu et al., 2017) further combines on-policy policy gradients with off-policy critics for stability and sample efficiency. For estimator selection, Udagawa et al. (2023) demonstrates adaptive, policy-aware estimator choice can be vital for robust performance. Dr. GRPO (Liu et al., 2025) proposes an unbiased alternative that eliminates these optimization biases and attains superior reasoning with fewer samples. To balance efficiency and adaptability in policy optimization, our approach integrates a static $V$ with a dynamic $Q$, ensuring stable advantage estimation and low computational overhead while maintaining responsive adaptation to policy updates.

## 3 PRELIMINARY

In this section, we review the fundamental concepts of policy optimization in RL, with a particular focus on the role of the advantage function and its various estimation methods.

### 3.1 PROXIMAL POLICY OPTIMIZATION

Actor-critic methods, such as PPO, train a critic network $V_\phi(s)$ to provide a low-variance estimate of the state-value function $V^\pi(s)$ of state $s$. The state-value function is used to compute the advantage at each time step $t$, typically via Generalized Advantage Estimation (GAE) (Schulman et al., 2015):

$$\hat{A}_t^{\text{GAE}} = \sum_{l=0}^{\infty} (\gamma\lambda)^l \delta_{t+l}, \quad \delta_t = r_t + \gamma V_\phi(s_{t+1}) - V_\phi(s_t), \tag{1}$$

where $\lambda$ is a hyper-parameter, $\delta_t$ is the temporal difference error at time step $t$, $r_t$ is the immediate reward received at time step $t$, $\gamma$ is the discount factor. PPO then optimizes a clipped surrogate objective to update the actor network in a stable manner:

$$\mathcal{J}^{\text{PPO}}(\theta) = \mathbb{E}_{q\sim P(D), o\sim \pi_{\theta_{\text{old}}}(O|q)} \left[ \min\left( r_t(\theta)\hat{A}_t^{\text{GAE}}, \text{clip}(r_t(\theta), 1-\epsilon, 1+\epsilon)\hat{A}_t^{\text{GAE}} \right) \right], \tag{2}$$

where $q$ are questions sampled from the dataset $D$, $o$ are outputs sampled from the old policy $\pi_{\text{old}}$, importance sampling ratio $r_t(\theta) = \frac{\pi_\theta(o_t|q,o_{<t})}{\pi_{\theta_{\text{old}}}(o_t|q,o_{<t})}$, $\epsilon$ is the clipping range of $r_t(\theta)$.

### 3.2 GROUP RELATIVE POLICY OPTIMIZATION

Since the critic network is typically as large as the actor network, it adds substantial memory and computational burden. Critic-free methods, such as GRPO, eliminate this costly component by estimating the advantage directly from rewards.

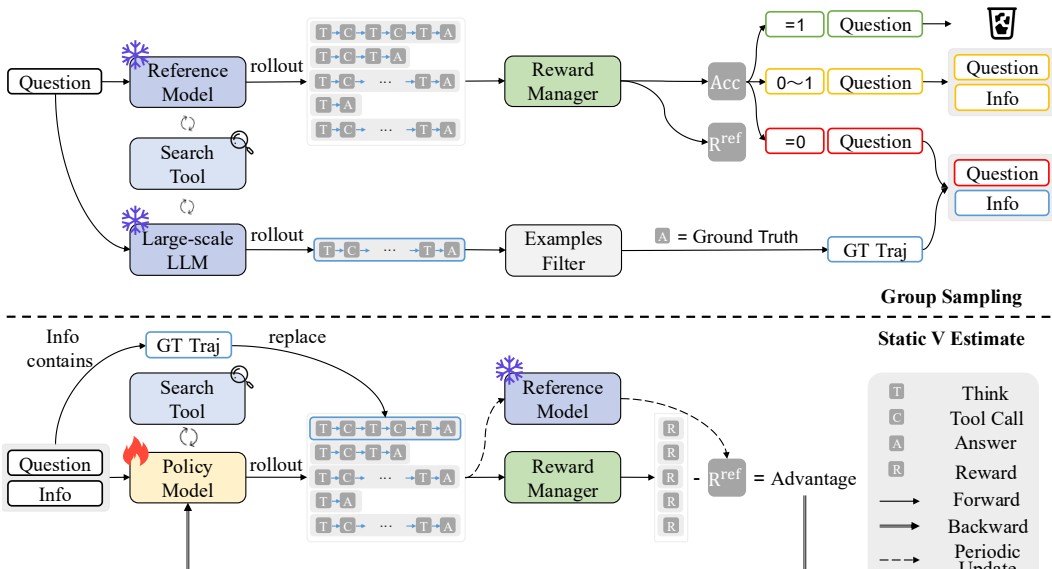

Figure 1: The architecture of PVPO. Reference model updates $R^{\text{ref}}$ at fixed steps, maintaining value stability and improving the performance lower bound. Reward manager do not restrict the generation of reward.

For each question, GRPO generates a group of outputs $\{o_i\}$ from the old policy $\pi_{\theta_{\text{old}}}$. The advantage for each output $o_i$ is then calculated based on normalized reward $\mathbf{r}$ relative to the group:

$$\hat{A}_{i,t} = \frac{r_i - \text{mean}(\mathbf{r})}{\text{std}(\mathbf{r})}. \tag{3}$$

This critic-free advantage estimate is then used to optimize a PPO-like objective function:

$$\mathcal{J}^{\text{GRPO}}(\theta) = \mathbb{E}_{q\sim P(D),\{o_i\}\sim\pi_{\theta_{\text{old}}}(O|q)}$$

$$\left[\frac{1}{G}\sum_{i=1}^{G}\frac{1}{|o_i|}\sum_{t=1}^{|o_i|}\left\{\min\left(r_{i,t}(\theta)\hat{A}_{i,t},\text{clip}\left(r_{i,t}(\theta),1-\epsilon,1+\epsilon\right)\hat{A}_{i,t}\right)-\beta D_{KL}[\pi_\theta||\pi_{\text{ref}}]\right\}\right], \tag{4}$$

where $r_{i,t}(\theta) = \frac{\pi_\theta(o_{i,t}|q,o_{i,<t})}{\pi_{\theta_{\text{old}}}(o_{i,t}|q,o_{i,<t})}$, $D_{KL}$ is the KL divergence between the trained policy $\pi_\theta$ and the reference policy $\pi_{\text{ref}}$, $\beta$ is a hyper-parameter.

## 4 METHODOLOGY

In this section, we will introduce our efficient and effective RL algorithm PVPO. The architecture is illustrated in Figure 1. PVPO optimizes the policy via the following objective:

$$\mathcal{J}^{\text{PVPO}}(\theta) = \mathbb{E}_{q\sim P(D),\{o_i\}\sim\pi_{\theta_{\text{old}}}(O|q)}$$

$$\left[\frac{1}{G}\sum_{i=1}^{G}\frac{1}{|o_i|}\sum_{t=1}^{|o_i|}\left\{\min\left(r_{i,t}(\theta)\hat{A}_{i,t}^{\text{PVPO}},\text{clip}\left(r_{i,t}(\theta),1-\epsilon,1+\epsilon\right)\hat{A}_{i,t}^{\text{PVPO}}\right)-\beta D_{KL}[\pi_\theta||\pi_{\text{ref}}]\right\}\right]. \tag{5}$$

where

$$r_{i,t}(\theta) = \begin{cases} \frac{\pi_\theta(o_{i,t}|q,o_{i,<t})}{\pi_{\theta_{\text{old}}}(o_{i,t}|q,o_{i,<t})}, & \text{if } o_i \notin \text{GT Traj.} \\ \frac{\pi_\theta(o_{i,t}|q,o_{i,<t})}{\pi_{\theta_{\text{gt}}}(o_{i,t}|q,o_{i,<t})}, & \text{if } o_i \in \text{GT Traj.} \end{cases} \tag{6}$$

## 4.1 STATIC V ESTIMATE

In actual policy optimization, the current method is to operate at the group level rather than through single sampling. For problem $q$, we use the current policy $\pi_\theta$ to generate $N$ independent trajectories $\mathcal{T} = \{\tau_1, \tau_2, ..., \tau_N\}$ and obtain the corresponding rewards $\mathbf{r} = \{R(\tau_1), R(\tau_2), ..., R(\tau_N)\} = \{r_1, r_2, ..., r_N\}$. For any step $(s_{i,t}, a_{i,t})$ in a specific trajectory $\tau_i$, the unbiased Monte Carlo estimate of the action value $Q^\pi(s_{i,t}, a_{i,t})$ is the final reward $r_i$ observed in that trajectory. We refer to this as the **Dynamic Q Estimate** because it directly reflects the result of a single rollout of the current policy:

$$\hat{Q}_{\text{dyn}}(\tau_i) = \mathbb{E}_{\tau \sim \pi_\theta}[R(\tau_i)] = r_i. \tag{7}$$

Considering that reward $r_i$ is given after the generation of trajectory $\tau_i$, the trajectory generation process is regarded as atomic actions $a_i = \tau_i$ executed from $s_{i,0}$. This atomicity makes the reward distribution of the intermediate state $s_{i,t}$ only depend on initial state $s_{i,0}$ ($s_0$) and $\pi_i$. Consequently, the expected return of the policy is equal to the state value of the initial state $V^\pi(s_0)$. A natural estimation method is to approximate this expectation using the empirical mean of all rewards in the current group. This is the approach adopted by on-policy methods such as GRPO, which we refer to as **Dynamic V Estimate**:

$$\hat{V}_{\text{dyn}}(s_0) = \hat{V}_{\text{dyn}}(\mathcal{T}) = \frac{1}{N} \sum_{j=1}^{N} r_j = \text{mean}(\mathbf{r}). \tag{8}$$

So we obtain the sparse advantage estimate for trajectory $\tau_i$ in the on-policy method:

$$\hat{A}_{\text{dyn}}(\tau_i, s_0) = \hat{Q}_{\text{dyn}}(\tau_i) - \hat{V}_{\text{dyn}}(s_0) = r_i - \text{mean}(\mathbf{r}). \tag{9}$$

This formula clearly shows that the advantage is calculated as the difference between the immediate reward and the average performance of the current policy $\pi_\theta$ within the group. However, $\hat{V}_{\text{dyn}}$ fluctuates wildly with each sampling of the group and is directly affected by $\pi_\theta$, introducing significant instability, especially when the group size is not large enough. To more effectively mitigate the instability associated with dynamic $V$ estimation, we propose substituting it with a more robust fixed $V$ estimate.

The ideal baseline should represent a **Reference Anchor** that does not change with current policy iterations. Therefore, we use the expected return of a fixed reference policy $\pi_{ref}$ (e.g., the initial policy model) as our **Static V Estimate** $\hat{V}_{\text{sta}}$. The baseline can be accurately estimated in advance by sampling the reference policy $\pi_{ref}$ M times, and update at fixed steps during training process:

$$\hat{V}_{\text{sta}}(s_0) = \frac{1}{M} \sum_{j=1}^{M} r_j^{\text{ref}} = \text{mean}(\mathbf{r}^{\text{ref}}). \tag{10}$$

This stable static baseline replaces the unstable dynamic baseline in formula 8. We finally obtain the advantage function of PVPO, which is well-suited for RL tasks with sparse rewards.

$$\hat{A}^{\text{PVPO}}(\tau_i, s_0) = \hat{Q}_{\text{dyn}}(\tau_i) - \hat{V}_{\text{sta}}(s_0) = r_i - \text{mean}(\mathbf{r}^{\text{ref}}). \tag{11}$$

In summary, our advantage function follows the original definition without further normalization, where $\hat{Q}_{\text{dyn}}(\tau_i)$ is obtained from the immediate reward of on-policy $\pi_\theta$ rollout. It reflects the current performance of the policy and is highly adaptive. The **Static V Estimate** $\hat{V}_{\text{sta}}(s_0)$ is obtained from the average reward of the reference policy $\pi_{\text{ref}}$ pre-rollout. It provides a stable and low-variance performance baseline.

## 4.2 GROUP SAMPLING

Inspired by DAPO's dynamic sampling strategy, we also assess the accuracy of sample rollouts while continuing to utilize the reference model for offline rollouts. For each sample, the mean accuracy of the rollouts serves as the filtering criterion.

Specifically, samples are categorized into three groups:

- Samples with a mean accuracy of 1 are excluded from the training set, as they are considered too trivial to facilitate effective learning.
- Samples with a mean accuracy strictly between 0 and 1 are retained, given their nonzero advantage.
- For samples exhibiting a mean accuracy of 0, an additional rollout is conducted using a larger LLM for further evaluation.

The larger LLM can correctly answer some of these samples. We cache these Ground Truth Trajectories (GT Traj) and their probability distributions. During policy training, a GT Traj is injected by replacing one of the generated rollouts for these specific samples. This method mitigates the sparse reward issue commonly encountered with complex samples. In the absence of guidance, the LLM may fail to obtain any positive feedback through unguided exploration. By providing a reference trajectory, the model receives an explicit demonstration, which jumpstarts learning by offering a clear example of a successful reasoning process.

## 5 EXPRIMENTS SETTING

**Metrics.** For multi-hop QA tasks, we employ answer accuracy (Acc, %) and LLM-as-a-Judge (LasJ, %) (Song et al., 2025) as evaluation metrics. For mathematical reasoning tasks, we measure answer accuracy (Acc, %), reporting the mean accuracy across 32 independent rollouts for each sample (i.e., acc@32).

**Datasets.** For multi-hop QA tasks, we conduct experiments on four multi-step retrieval datasets: Musique (Trivedi et al., 2022), 2WikiMultiHopQA (2Wiki) (Ho et al., 2020), HotpotQA (Yang et al., 2018), and Bamboogle (Bam) (Press et al., 2023). Model training is performed on the Musique training split, which consists of 20k examples, and evaluations are carried out on the full development and test sets. For mathematical reasoning tasks, we train models on DAPO-Math-17k-Processed (Yu et al., 2025), comprising 17k examples, and conduct evaluation on five test sets: DAPO-AIME-2024 (AI-MO, 2024; Bytedance & Tsinghua-SIA, 2025), AIME-2025 (Lin, 2025), MATH500 (Lightman et al., 2024; HuggingFaceH4, 2023), AMC23 (AI-MO, 2024), and Olympiad (He et al., 2024).

**Baselines and Training Details.** We use *Qwen2.5-7B-Instruct* and *Qwen2.5-14B-Instruct* as base models and *Qwen2.5-72B-Instruct* as the large LLM to generate GT Traj. The reference reward $R^{\text{ref}}$ is updated every 500 steps. For training, we set the learning rate to 1e-6, maximum response length to 8192, sampling temperature to 1.0 and top-p to 1.0. For inference, we set the sampling temperature to 0.6 and top-p to 0.95. For the multi-hop QA tasks, we benchmark our method against not only state-of-the-art LLMs such as *DeepSeek-R1-0528*, *GPT-4.1-0414*, *O4-mini-0416*, and *Gemini-2.5-pro-0325*, but also prominent RL-based agentic search models (Jin et al., 2025; Song et al., 2025). We adopt the ReSearch (Chen et al., 2025) framework, with pre-samples $M = 5$, rollout $N = 5$, train batch size of 8, and 1,000 training steps. For DynaSearcher(Hao et al., 2025), we remove the "kg_filter" during inference. For mathematical reasoning tasks, we primarily adopt GRPO (Shao et al., 2024), DAPO (Yu et al., 2025), and GSPO (Zheng et al., 2025) as baselines. We use the verl (Sheng et al., 2025) framework with pre-samples $M = 16$, rollout $N = 16$, train batch size of 32, and 1,000 training steps. For DAPO, we set the clipping parameter $\epsilon_{\text{low}} = 0.2$ and $\epsilon_{\text{high}} = 0.28$. For GRPO, we set the "loss_agg_mode" to "seq-mean-token-mean", which is aligned with the original paper. For GSPO, the clipping parameter $\epsilon$ is set to 0.0003. All experiments are conducted on a server equipped with an Intel(R) Xeon(R) Platinum 8369B CPU and $8\times$NVIDIA A100-SXM4-80GB GPUs. More details can be found in Appendix A.1.

## 6 EXPERIMENTS

In this section, we conduct a series of experiments to comprehensively evaluate PVPO. First, we test our method on multi-hop QA to validate its effectiveness in the agent domain. Next, we perform ablation studies to examine the contributions of the core modules of PVPO. We further apply PVPO to mathematical reasoning tasks to verify its generalizability and also evaluate its compatibility with other advanced RL algorithms. In addition, we analyze the training efficiency and convergence properties of PVPO. Finally, we present a case study to investigate the efficiency and robustness of PVPO under low sampling budget.

Table 1: Performance comparisons between PVPO and the baselines on multi-step retrieval datasets. The best and second best results are **bold** and underlined, respectively.

| Method | Musique | | 2Wiki | | HotpotQA | | Bamboogle | | Average | |
|---|---|---|---|---|---|---|---|---|---|---|
| | Acc | LasJ | Acc | LasJ | Acc | LasJ | Acc | LasJ | Acc | LasJ |
| *Prompt Based* | | | | | | | | | | |
| Qwen2.5-7B-Instruct | 5.1 | 13.5 | 27.9 | 29.3 | 22.4 | 31.0 | 12.8 | 17.1 | 17.1 | 22.7 |
| DeepSeek-R1 | 32.0 | 40.7 | 57.5 | 59.4 | 43.0 | 58.3 | 66.4 | 76.6 | 49.7 | 58.8 |
| O4-mini | 38.0 | 44.1 | 61.5 | 67.4 | 49.5 | 67.4 | 74.4 | 84.2 | 55.9 | 65.8 |
| GPT-4.1-global | 31.0 | 40.9 | 58.0 | 58.5 | 44.5 | 57.7 | 51.2 | 61.6 | 46.2 | 54.7 |
| Gemini-2.5-pro | 42.5 | 50.8 | 70.0 | 71.2 | 53.0 | 71.1 | **75.2** | **84.5** | 60.2 | 69.4 |
| *Train Based* | | | | | | | | | | |
| *Qwen2.5-7B-Instruct* | | | | | | | | | | |
| Search-R1-v0.3 | 24.7 | 34.6 | 58.7 | 61.1 | 53.6 | 66.9 | 48.0 | 54.5 | 46.3 | 54.4 |
| R1-Searcher | 24.7 | 34.2 | 67.8 | 68.2 | 59.7 | 71.5 | 46.4 | 52.0 | 50.5 | 56.5 |
| GRPO-ReSearch | 33.4 | 46.7 | 60.8 | 67.0 | 54.5 | 63.7 | 45.6 | 54.4 | 48.6 | 58.0 |
| GRPO-DynaSearcher | 38.9 | 52.0 | 74.3 | 76.8 | 62.7 | 68.3 | 51.2 | 58.7 | 56.8 | 64.0 |
| PVPO-ReSearch | 36.5 | 51.4 | 70.1 | 72.4 | 65.5 | 72.3 | 45.6 | 54.3 | 54.4 | 62.6 |
| PVPO-DynaSearcher | **46.9** | **59.4** | **77.7** | **80.6** | **69.0** | **78.4** | 50.4 | 59.7 | **61.0** | **69.6** |

## 6.1 MAIN RESULTS

We evaluate PVPO against both zero-shot leading LLMs and trained RL-based search methods, with results in Table 1 underscoring its effectiveness. Specifically, applying PVPO substantially improves the base frameworks, boosting ReSearch's Avg Acc/LasJ scores by 5.8/4.6 points and DynaSearcher's by 4.2/5.6 points. Notably, our PVPO-DynaSearcher model significantly outperforms all RL-trained baselines (e.g., surpassing GRPO by over 5 points on average). It also marginally exceeding the strongest proprietary LLM, *Gemini-2.5-Pro*, while establishing a considerable lead over other models like *O4-mini*, *GPT-4.1*, and *DeepSeek-R1*. On the Bamboogle dataset, SOTA LLMs significantly outperform 7B-trained models largely due to the outdated 2018 Wikipedia corpus used in our experiments (see Appendix A.1 and Figure 5). Overall, these results demonstrate that PVPO consistently achieves state-of-the-art performance across agentic search methods.

## 6.2 ABLATION STUDY

We conduct an ablation study to isolate the contribution of each component in PVPO, as shown in Table 2. Starting from the GRPO-DynaSearcher baseline (56.8 Avg Acc / 64.0 LasJ), the integration of Static V Estimation first raises the scores to 58.3/66.7. Subsequently adding Group Sampling further boosts the performance to 61.0/69.6, which represents our full PVPO model and outperforms all baselines. This incremental improvement validates the effectiveness of each proposed component.

Table 2: Ablation study of PVPO on multi-step retrieval datasets. Starting from DynaSearcher on Qwen2.5-7B-Instruct, we incrementally add Static V Estimation and Group Sampling.

| Method | Average | |
|---|---|---|
| | Acc | LasJ |
| Qwen2.5-7B-Instruct | 31.1 | 39.7 |
| GRPO-DynaSearcher | 56.8 | 64.0 |
| + Static V Estimation | 58.3 | 66.7 |
| + Group Sampling | 61.0 | 69.6 |

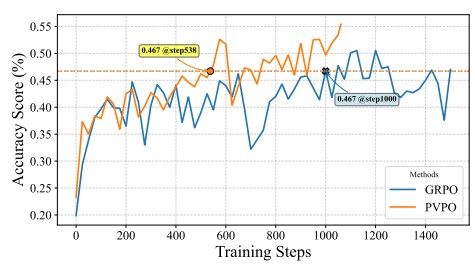

Figure 2: Training efficiency of PVPO on mathematical reasoning datasets.

Table 3: Performance comparison of PVPO and baseline methods on mathematical reasoning datasets using different model scales. "w/" means trained with.

| Method | MATH500 | AMC23 | Olympiad | AIME-2024 | AIME-2025 | Avg Acc |
|---|---|---|---|---|---|---|
| Qwen2.5-7B-Instruct | 75.68 | 42.92 | 38.94 | 12.10 | 6.67 | 35.26 |
| w/ GRPO | 78.60 | 49.10 | 42.14 | 13.86 | 10.10 | 38.76 |
| w/ DAPO | 78.58 | 51.38 | 43.36 | 14.96 | 11.30 | 39.92 |
| w/ GSPO | 78.66 | 50.12 | 43.60 | 15.02 | 12.70 | 40.02 |
| w/ PVPO | **80.30** | **52.02** | **44.62** | **14.86** | **14.70** | **41.30** |
| Qwen2.5-14B-Instruct | 79.68 | 51.52 | 44.00 | 14.82 | 12.29 | 40.46 |
| w/ GRPO | 82.12 | 53.50 | 47.42 | 16.14 | 15.86 | 43.01 |
| w/ DAPO | 82.50 | 56.44 | 49.34 | 18.04 | 15.66 | 44.40 |
| w/ GSPO | 83.56 | 56.02 | 49.28 | 18.18 | 16.20 | 44.65 |
| w/ PVPO | **83.64** | **56.78** | **50.72** | **19.24** | **17.74** | **45.62** |

## 6.3 GENERALIZATION EVALUATION

To evaluate the transferability of PVPO, we apply it to mathematical reasoning tasks spanning a range of difficulties, from basic arithmetic to olympiad-level problems. We compare PVPO with GRPO, DAPO, and GSPO across several benchmark datasets. As shown in Table 3, PVPO consistently outperforms all baselines on both the 7B and 14B model scales. We further combine PVPO with the core modules of advanced RL methods, such as the sequence-level importance ratio from GSPO and the KL removal strategy from DAPO, achieving additional performance improvements when integrated with these state-of-the-art algorithms. Since these integrated modules are not the main focus of PVPO, we provide the detailed results and metrics for these extensions in Appendix A.3 and Table 4. Furthermore, PVPO exhibits robust cross-domain generalization and enhanced scalability.

## 6.4 TRAINING EFFICIENCY ANALYSIS

As illustrated in Figure 2, PVPO converges much faster than GRPO, reaching the target accuracy in only 500 steps compared to GRPO's 1,000 steps. After 1,000 steps, PVPO also achieves higher final accuracy, confirming its effectiveness. By applying Group Sampling, PVPO filters out 40–60% of low-quality data and further accelerates training by $1.7\times$ to $2.5\times$ (see Appendix A.2). Overall, these results confirm that PVPO improves both convergence speed and training efficiency.

## 6.5 STABILITY EVALUATION

We track PVPO training metrics to show its stability. Figure 3 (a) shows that PVPO achieves a much higher average reward than GRPO. With a similar KL divergence in Figure 3 (b), this improvement comes not from more aggressive updates, but from better gradient direction estimates. As shown in Figure 3 (c), PVPO has lower advantage variance, leading to more reliable and consistent update directions. PVPO also maintains exploration without losing stability. Figure 3 (d) shows that it keeps higher policy entropy under a similar KL constraint, which helps avoid premature convergence to a local optimum. Overall, PVPO addresses key problems in RL by supporting high exploration, low variance, and high rewards, thereby achieving more stable training than existing methods.

## 6.6 CASE STUDY: LOW SAMPLING BUDGET

To further examine PVPO's performance under resource constraints, we conduct a case study on low sampling budget. We reduce the number of rollouts from 5 (used in the main experiments) to 2. For comparison, we report GRPO's performance with a full budget. Figure 4 (a) shows that PVPO with a low budget remains close to the fully budgeted GRPO. We calculate computational cost by multiplying the number of rollouts with the average number of tool calls in trajectories. As shown in Figure 4 (b), PVPO's average cost is only 4.3, which is much lower than GRPO's 11.7. PVPO achieves 97% of GRPO's performance (55.0% vs 56.8%) while using less than 40% of the computational cost. This strong sample efficiency comes from the high-quality, low-variance

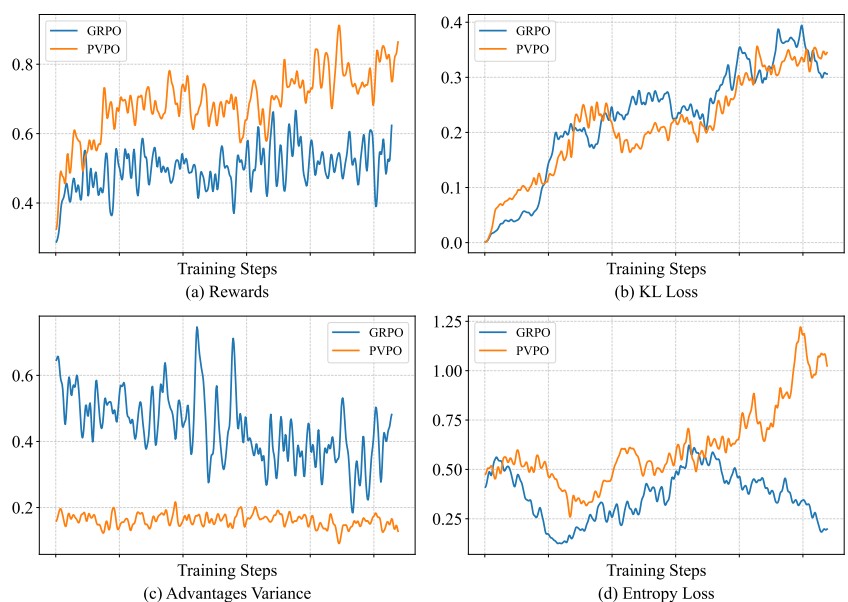

Figure 3: Training stability of PVPO on multi-step retrieval datasets.

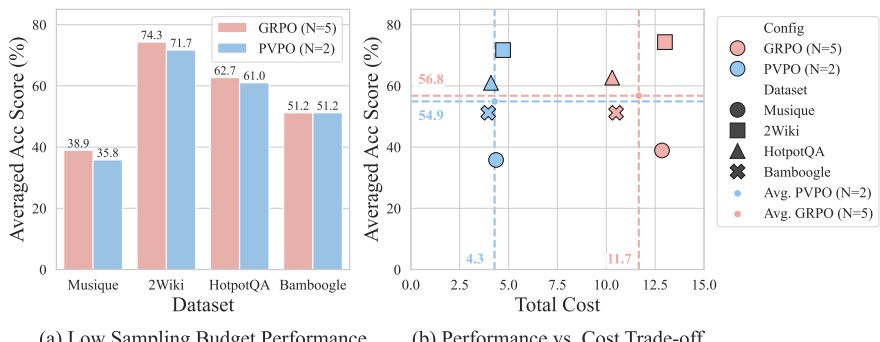

Figure 4: Low sampling budget of PVPO on multi-step retrieval datasets. The $N$ denotes the number of trajectories in each single rollout. $N$=5 is the full budget and $N$=2 is the low budget.

training signals provided by Static V Estimate. The model can update its policy efficiently using fewer rollouts.

## 7 CONCLUSIONS

In this paper, we propose PVPO, an efficient critic-free reinforcement learning algorithm designed to optimize policy learning for complex tasks. By introducing a Static V Estimate as an external advantage reference and integrating it with group sampling for effective data filtering, PVPO addresses the limitations of extensive sampling and biased intra-group comparisons inherent in prior methods. Our approach yields stable, low-variance training signals, accelerates convergence, and significantly reduces computational costs. Extensive experiments across nine diverse benchmarks in multi-hop QA and mathematical reasoning demonstrate that PVPO achieves state-of-the-art performance and strong generalization, even with small-scale models and limited resources. PVPO introduces substantial improvements in reasoning and tool use, supports scalable training, and ensures consistent performance, thereby demonstrating strong potential for widespread real-world application.

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

# A APPENDIX

## A.1 IMPLEMENTATION DETAILS

**Retriever and Corpus**. For the multi-hop QA task, we employ *multilingual-e5-base* as the retriever model and use the December 2018 Wikipedia dump as the primary retrieval corpus, which contains over 21 million passages. To improve retrieval efficiency, we construct the final corpus by combining supporting document passages from three multi-hop datasets (i.e., Musique, 2Wiki, and HotpotQA) with one million randomly sampled documents from the Wikipedia dump. Notably, Bamboogle only provides questions and answers without ground truth passages, so it cannot be incorporated into the retrieval corpus. This may contribute to the lower scores on Bamboogle for most methods, as shown in Table 1.Passage retrieval is implemented using FAISS[1], and for each query, the top 5 passages are retrieved during both training and testing. For the KG (Knowledge Graph) data used in **PVPO-DynaSearcher**, we follow the approach and dataset provided by Wang et al. (2021), which is aligned with Hao et al. (2025).

**Prompts and Code**. We implement **PVPO-ReSearch** and **PVPO-DynaSearcher** based on the ReSearch framework[2]. The system prompts for ReSearch and DynaSearcher are set following their respective original papers, detailed prompt templates are shown in Figure 6 and 7. For prompt-based SOTA LLMs, we first retrieve 5 passages from the corpus for each question, and then organize these passages using the template shown in Figure 5 as the prompt for answer generation. For mathematical reasoning tasks, we use `verl` version `0.3.1.dev0`. Since the ReSearch codebase is also developed on top of the verl framework, we provide the core implementation of our PVPO method based on verl in code Listing 1 and Algorithm 1.

```python
# verl/trainer/ppo/ray_trainer.py
def compute_advantage(...):
    if adv_estimator == AdvantageEstimator.PVPO:
        # compute pvpo advantages
        advantages, returns = core_algos.
            compute_pvpo_outcome_advantage(
            token_level_rewards=data.batch["token_level_rewards"],
            token_level_values=data.non_tensor_batch["static_value"
                ],
            response_mask=data.batch["response_mask"],
        )
        data.batch["advantages"] = advantages
        data.batch["returns"] = returns
    ...
# verl/trainer/ppo/core_algos.py
def compute_pvpo_outcome_advantage(
    token_level_rewards: torch.Tensor,
    token_level_values: torch.Tensor,
    response_mask: torch.Tensor,
):
    scores = token_level_rewards.sum(dim=-1)
    values = torch.tensor(token_level_values.astype(np.float32),
        device=scores.device, dtype=scores.dtype)

    with torch.no_grad():
        for i in range(scores.shape[0]):
            scores[i] = (scores[i] - values[i])
        scores = scores.unsqueeze(-1) * response_mask
    return scores
```

Listing 1: PyTorch-style pseudocode for PVPO

---

---

**Algorithm 1** Replace Incorrect Rollout with Ground Truth for Zero-Accuracy Prompts

---

1: **function** REPLACEINCORRECTROLLOUTSWITHGT(batch, reward_tensor, tokenizer, ...)
2:     **if** PVPO advantage estimator is enabled **then**
3:         acc_tensor ← Compute accuracy for each rollout
4:         indices ← Find indices where all rollouts are incorrect        ▷ Accuracy is zero
5:         **for** each prompt_idx in indices **do**
6:             REPLACE( prompt_uid, batch, reward_tensor, tokenizer, ...)
7:         **end for**
8:     **end if**
9: **end function**

10: **function** REPLACE(prompt_uid, batch, reward_tensor, tokenizer, ...)
11:     row ← Find first rollout in batch matching prompt_uid
12:     Retrieve ground truth: response, tokens, log_probs, mask from batch
13:     **if** ground truth trajectory is missing **or** gt response length > max_response_length **then**
14:         **return** batch, reward_tensor
15:     **end if**
16:     Update batch with GT: responses, log_probs, mask at row
17:     Reconstruct input_ids, attention_mask, position_ids to align batch
18:     Set reward on last valid token of GT to 1
19:     **return** batch, reward_tensor
20: **end function**

---

You are an expert in question answering. Given a question within `<question>` `</question>` and some contexts within `<context>` `</context>`, you first think about the reasoning process within `<think>` `</think>` and put the answer within `<answer>` `</answer>`. For example, ¡question¿ This is a question `<question>` `<context>` Here are contexts `<context>` `<think>` This is the reasoning process. `</think>` `<answer>` The final answer is \boxed{ answer here } `</answer>`. If the answer could not be deduced from the contexts or it's wrong, give the right answer based on your own knowledge. In the last part of the answer, the final exact answer is enclosed within \boxed{}.

Figure 5: Prompt for zero-shot LLM RAG.

You are a helpful assistant that can solve the given question step by step with the help of the wikipedia search tool. Given a question, you need to first think about the reasoning process in the mind and then provide the answer. During thinking, you can invoke the wikipedia search tool to search for fact information about specific topics if needed. The reasoning process and answer are enclosed within `<think>` `</think>` and `<answer>` `</answer>` tags respectively, and the search query and result are enclosed within `<search>` `</search>` and `<result>` `</result>` tags respectively. For example, `<think>` This is the reasoning process. `</think>` `<search>` search query here `</search>` `<result>` search result here `</result>` `<think>` This is the reasoning process. `</think>` `<answer>` The final answer is \boxed{ answer here } `</answer>`. In the last part of the answer, the final exact answer is enclosed within \boxed{}.

Figure 6: System prompt for ReSearch.

You are a helpful assistant that can solve the given question step by step with the help of the wikipedia search tool. Given a question, you need to first think about the reasoning process in the mind and then provide the answer. During thinking, you can invoke the wikipedia search tool to search for fact information about specific topics if needed. The reasoning process and answer are enclosed within <think> </think> and <answer> </answer> tags respectively, and the search input and result are enclosed within <search> </search> and <result> </result> tags respectively. Search input is json format like {"query": "xxx", "entity": ["yyy"], "relation": ["zzz"]} and applied to the search tools, where query is used to search wikipedia articles, entity(s) and relation(s) are used to search wikidata, a knowledge base of entities and relations.
For example, <think> This is the reasoning process. </think> <search> {"query": "Who is the director of Avatar", "entity": ["Avatar"], "relation": ["director"]} </search> <result> search result here </result> <think> This is the reasoning process. </think> <answer> The final answer is \boxed{ answer here }</answer>. In the last part of the answer, the final exact answer is enclosed within \boxed{}.

Figure 7: System prompt for DynaSearcher.

You will be provided with three pieces of content: the questioner's question, the user's response, and the reference answer list. Your task is to score the accuracy of the user's response based on the criteria outlined below. Please ensure that you carefully read and understand these instructions. Evaluation Criteria: 1. The pred answer doesn't need to be exactly the same as any of the ground truth answers, but should be semantically same for the question. 2. Each item in the ground truth answer list can be viewed as a ground truth answer for the question, and the pred answer should be semantically same to at least one of them. 3. The user's response may be longer and more detailed; as long as it is logically correct, contains the correct answer, it should be scored appropriately. Evaluation Steps: 1. Carefully read the questioner's question and understand its key points. 2. Carefully read the reference answer and understand the key points relevant to the question. 3. Based on the evaluation criteria, assign a score in the range of 0 to 5, where 0 indicates that the user's response does not include any of the key points from the reference answer and completely fails to answer the questioner's question; 5 indicates that the user's response includes all the key points from the reference answer and fully and correctly answers the questioner's question.
Questioner's question: {question}
Reference answer: {answer}
User's response: {response}
Evaluation result (output only the score between 0 and 5):

Figure 8: Prompt for LLM-as-Judge score.

## A.2 GROUP SAMPLING ANALYSIS

We calculate the data filtering ratio on two training sets, as shown in Figure 9. Group Sampling removes samples with Acc = 1 or 0 before training, filtering out 40%-60% of the total dataset. This leads to a 1.7–2.5× increase in training efficiency.

## A.3 ADDITIONAL EXPERIMENT RESULTS

To further verify the scalability of our proposed PVPO method, we conduct integration experiments on multi-hop QA tasks. Specifically, we combine PVPO with the sequence-level importance ratio module proposed in GSPO and remove the KL loss constraint as introduced in DAPO. The results, shown in Table 4, demonstrate that PVPO not only provides strong baseline improvements over GRPO, but also achieves further performance gains when integrated with these advanced RL meth-

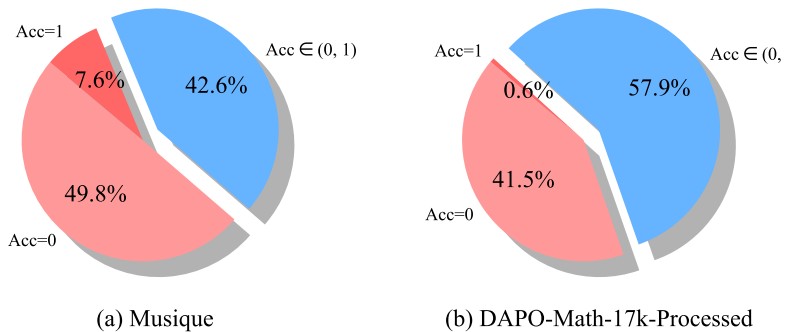

(a) Musique          (b) DAPO-Math-17k-Processed

Figure 9: Group Sampling study on datasets from different fields. The Acc is the mean of the answer accuracies from M trajectories rolled out by the reference model. M=5 in Figure (a) and M=16 in Figure (b).

Table 4: Experimental results of PVPO's orthogonal integration with SOTA RL algorithms (DAPO, GSPO) and scalability evaluation on multi-hop QA tasks. "w/ Seq-Ratio" refers to the sequence-level importance ratio from GSPO, and "w/o KL" means removing the KL loss constraint as in DAPO.

| Method | Average | | |
|---|---|---|---|
| | Acc | LasJ | ToolCalls |
| GRPO-ReSearch | 48.6 | 58.0 | 2.46 |
| PVPO-ReSearch | 54.4 | 62.6 | 2.96 |
| w/ Seq-Ratio (GSPO) | 55.1 | 62.4 | 2.19 |
| w/o KL (DAPO) | **58.8** | **67.1** | 8.14 |

ods. In particular, the combination with DAPO (w/o KL) yields the best accuracy and LasJ scores, while integration with GSPO's sequence-level importance ratio also presents consistent improvements. In particular, the combination with DAPO (w/o KL) yields the best accuracy and LasJ scores, but also incurs significantly more tool calls (8.14 per query), resulting in greater inference costs. By contrast, GSPO's sequence-level importance ratio offers improvements with relatively lower tool call overhead (2.19 per query). Therefore, the trade-off between performance and inference cost should be considered when choosing an integration strategy for different practical scenarios. These findings confirm that PVPO is highly compatible and complementary when used alongside other state-of-the-art RL algorithms.

