# OpenReview forum: "PVPO: Pre-Estimated Value-Based Policy Optimization for Agentic Reasoning"
_ICLR.cc/2026/Conference — ICLR 2026 Conference Withdrawn Submission_

### Official Review · Reviewer_WdaM · 2025-10-27

**Soundness:** 1
**Presentation:** 2
**Contribution:** 2
**Rating:** 2
**Confidence:** 4

**Summary:**

The authors propose an critic-free RL method for large model training that offers an alternative to group policies which require multiple samples per query.  Their method avoids group-based advantage estimation by using a static value function instead, which is just the average reward over multiple rollouts of the reference policy, and can be pre-computed.  They also generate synthetic demonstrations by using a larger LLM to generate ground truth trajectories for samples with 0 mean accuracy.  The compare with both prompt-based LLMs and GRPO on multi-hop QA tasks and reasoning tasks with strong performance overall.

**Strengths:**

1. The authors identify a sample efficiency problem in critic free RL and propose a reasonable method of using a static value baseline.
2. The authors demonstrate strong performance over multiple benchmarks.

**Weaknesses:**

1. Comparisons may be unfair.  Their method uses a larger LLM to generate trajectories for difficult tasks which makes it an unfair comparison to GRPO which does not.  Do you apply the same group sampling to other comparison methods?  If not, PVPO is getting privileged information which explains its strong performance.

2. The sample efficiency/cost gains of PVPO are not clear.  One of the main claims is that this method is a more efficient RL method because it bypasses multiple sampling.  However, the static value estimates still need to be updated periodically and it requires additional trajectories from a larger LLM.  How does the actual combined sample efficiency compare?

3. There are multiple unsupported or wrong statements in the writing.
- Abstract states "our approach effectively corrects the cumulative bias introduced by intra-group comparisons".  This is never explained later in the paper.
- Intro states "effectively mitigating concerns of error accumulation and policy drift during training".  Nothing in the method explicitly addresses error accumulation or policy drift in comparison to baseline methods.
- 4.1 "estimate of the action value Q is the final reward r_i observed in the trajectory".  This is true only if there are no intermediate rewards in the episode and the discount is 1.
- Training efficiency results in Figure 2 aren't conclusive.  This is a high variance learning curve over what looks to be a single seed.

**Questions:**

1. Section 4.1 "the trajectory generation process is regarded as atomic actions a_i=tau_i [...] makes the reward distribution [..] depend only on initial state and pi".  Does this mean that the entire action sequence is generated at once from $$s_{i,0}$$.  If so, how do you handle the responses from the multi-hop QA problems?
2. Do you regenerate the static V estimate for all queries at a fixed interval throughout training?  If so, how does the number of these samples compare to the group samples used in GRPO for advantage estimation?  Are these samples used for PPO updates?  Does your method break down into GRPO with reward caching?

---

### Official Review · Reviewer_xtLf · 2025-10-29

**Soundness:** 2
**Presentation:** 3
**Contribution:** 2
**Rating:** 4
**Confidence:** 4

**Summary:**

The paper proposes PVPO, a critic-free RL method for agentic reasoning. It replaces GRPO’s group-wise “dynamic V” baseline with a Static-V baseline using a fixed reference policy. It further introduces Group Sampling to filter trivial items and  inject  Ground-Truth Trajectories (GT Traj) generated by a larger LLM for difficult items.  On multi-hop retrieval (ReSearch/DynaSearcher) and math reasoning (7B/14B), PVPO outperforms GRPO-family baselines and converges faster.

**Strengths:**

* Addresses a critical issue—the high variance in GRPO—which is important and timely to study.
* The use of GT trajectories for hard cases is well-motivated and methodologically sound.
* The method demonstrates significant, consistent improvements over strong baselines.

**Weaknesses:**

* Using a state-independent/reference baseline for variance reduction is well established in policy-gradient and recent critic-free methods. The paper should more systematically position the “Static-V anchor” relative to Dr.GRPO, DAPO, and GSPO, clarifying its distinct contribution and the conditions under which it outperforms these methods.
* The evidence is predominantly empirical; formal guarantees (e.g., convergence or improvement bounds, bias/variance analysis) are missing.

* The observed stability appears to come from two design choices: (i) a fixed reference model and (ii) a cached baseline. An ablation of GRPO with a cached baseline (without group normalization)—matching PVPO’s refresh frequency and offline rollout count—would help isolate these effects.

**Questions:**

* Using a larger LLM to generate GT trajectories can introduce distribution shift. Is it necessary to use a larger model from the same family as the trained small model (as in the paper’s Qwen→Qwen setup)? What happens if the demonstrator comes from a different family (e.g., DeepSeek)?

* Why was the “kg filter” removed in DynaSearcher? How does its removal affect the relative gap between GRPO and PVPO? Please provide controlled experiments with the filter kept to isolate this effect.

---

### Official Review · Reviewer_x8Ph · 2025-10-31

**Soundness:** 2
**Presentation:** 2
**Contribution:** 2
**Rating:** 4
**Confidence:** 4

**Summary:**

This paper describes a novel RL method using reference anchors and data pre-sampling.

**Strengths:**

The research problem is interesting and useful.

**Weaknesses:**

1. The introduction should be improved. The research motivation and problem setting (e.g., sparse reward) are not very clear, and the keyword “agentic reasoning” in the title is never mentioned.

2. The writing of the technical part could and should be improved, too. For example, over 50% of Sec. 4.1 should be removed to Sec. 3.

3. I understand that the proposed static V estimation is stable, but why is it good enough, or how is a good reference policy determined?

**Questions:**

See detailed comments.

---

### Official Review · Reviewer_kihm · 2025-11-01

**Soundness:** 3
**Presentation:** 2
**Contribution:** 2
**Rating:** 4
**Confidence:** 4

**Summary:**

This paper argues that dynamic value estimates, as traditionally used in GRPO setting, leads to high variance estimate and local optimum. To overcome this, they propose to use static value estimate from a reference model. They also dynamically group samples based on the quality of the current response. They perform considerable number of experiments across nine different datasets to validate their approach.

**Strengths:**

### Strengths:

1. This work presents an alternative approach for critic-free RL that leverages a low-variance and globally consistent advantage function to address error accumulation and policy drift during training.

2. Validation and ablations are conducted across a good number of different datasets and tasks. These experimental results reflect the efficacy of the proposed approach.

**Weaknesses:**

### Weaknesses:

1. GRPO’s one of the main objectives is to reduce resource consumption. However, the proposed method needs to maintain a reference model and also uses a large model for ground truth in case of failure. This is somewhat contradictory and limits the overall gain.

2. Further, the memory overhead due to these additional components has not been discussed. Also, I would like to see some discussion on the limitations.

3. The details of the reference model is not clear. The paper mentions that the reference anchor is computed in an unsupervised manner, however, I do not see any detailed discussion on this. Are you updating the reference model after certain steps like a target network?

4. I would consider the update frequency of the reference reward as a hyperparameter. How is the value of that hyperparameter selected? How sensitive the model is to that hyperparameter?

**Questions:**

1. In line 76-77, what did you mean by spatio-temporal overhead?

Also, look at the weaknesses section for other questions.

---

### Note · Authors · 2025-11-25

I have read and agree with the venue's withdrawal policy on behalf of myself and my co-authors.